# Exercise in Diabetic Nephropathy: Protective Effects and Molecular Mechanism

**DOI:** 10.3390/ijms25073605

**Published:** 2024-03-23

**Authors:** Ruo-Ying Li, Liang Guo

**Affiliations:** 1School of Exercise and Health, Collaborative Innovation Center for Sports and Public Health, Shanghai University of Sport, Shanghai 200438, China; lry13218026536@163.com; 2Shanghai Frontiers Science Research Base of Exercise and Metabolic Health, Shanghai University of Sport, Shanghai 200438, China; 3Key Laboratory of Exercise and Health Sciences of the Ministry of Education, Shanghai University of Sport, Shanghai 200438, China

**Keywords:** diabetic nephropathy, exercise, AMPK, Sirt1, renin-angiotensin system, MicroRNAs, nitric oxide, heat shock protein, exerkine, metabolites, gut microbiota

## Abstract

Diabetic nephropathy (DN) is a serious complication of diabetes, and its progression is influenced by factors like oxidative stress, inflammation, cell death, and fibrosis. Compared to drug treatment, exercise offers a cost-effective and low-risk approach to slowing down DN progression. Through multiple ways and mechanisms, exercise helps to control blood sugar and blood pressure and reduce serum creatinine and albuminuria, thereby alleviating kidney damage. This review explores the beneficial effects of exercise on DN improvement and highlights its potential mechanisms for ameliorating DN. In-depth understanding of the role and mechanism of exercise in improving DN would pave the way for formulating safe and effective exercise programs for the treatment and prevention of DN.

## 1. Introduction

Diabetes is a prominent global contributor to end-stage renal disease (ESRD), which represents the advanced stage of kidney disease. Roughly one in three adults with diabetes develops chronic kidney disease (CKD), and this proportion is continually rising [1]. Diabetic nephropathy (DN) is a chronic disease marked by the gradual elevation of blood pressure, urinary albumin excretion, and cardiovascular risk. It is also associated with a decreased glomerular filtration rate (GFR) and ultimately leads to the progression to end-stage renal disease (ESRD) [2]. The mortality rate for individuals with DN is roughly 30 times higher compared to diabetic patients without kidney damage [3]. Thus, how to intervene in the early stages of DN to delay its progression to ESRD has become a very critical issue.

The management of patients with DN can be categorized into four primary aspects: mitigating cardiovascular risk, regulating glycemic levels, controlling blood pressure, and inhibiting the renin–angiotensin system (RAS) [4]. These interventions contribute to the improvement of DN. In addition to pharmaceutical interventions, exercise, especially aerobic exercises like brisk walking or jogging, is firmly established as an effective approach for preventing and managing cardiovascular disease, making it beneficial for slowing the advancement of DN [5]. Engaging in moderate physical activity (a minimum of two times per week) has been linked to a reduced risk of adverse renal outcomes and a decreased incidence of albuminuria [6]. A meta-analysis by Cai et al. [7] indicated that exercise can slow down the progression of DN. In addition, a number of studies have investigated the effects of different exercise intensities on DN, concluding that patients with DN received the greatest benefit from moderate-intensity exercise. Meanwhile, some studies using diabetic mice or rats as research subjects have confirmed that exercise can reduce the level of advanced glycation end products (AGEs) and ameliorate glomerular sclerosis in the superficial, intermediate, and proximal medullary areas of the renal cortex and damage in the surface and intermediate layers of the renal cortex [8].

The primary initiating pathogenesis in DN is hyperglycemia-induced vascular dysfunction, and the progression of DN is due to different pathological factors, including oxidative stress, inflammation, cell death, and fibrosis [9]. It is possible that exercise may improve DN through alleviating the effects of the above factors. Although the last decade has seen many studies unravelling that exercise is linked to health benefits for the kidneys in diabetic patients, the mechanism by which cellular, molecular, and biochemical pathways are affected and regulated by exercise is still without definite conclusion. Therefore, it is worthwhile to summarize the latest research progress on the role and mechanism of exercise-mediated improvement in DN in order to inspire new ideas. Many medical and non-medical therapies are aimed at preventing the complications caused by diabetes. But drug treatments have varying degrees of side effects, making non-drug treatments more attractive because of their fewer side effects. Exercise, as a major non-medical strategy, can reduce the development of diabetes complications, including DN [10].

In this review, we outline the importance of exercise in delaying DN progression, the correlation between patient-self-reported exercise frequency and DN, and the effects of different forms of exercise on DN. We also summarize and discuss the molecular mechanism of exercise-mediated improvement in DN, which may provide more insights for future research in this area.

## 2. Protective Role of Exercise against DN in Human Studies

Many human studies have indicated that exercise can delay the progression of DN. A meta-analysis [7] that included 38,991 participants showed that physical activity increased the GFR and decreased the urinary albumin creatinine ratio (UACR) in patients with diabetes, and physical activity reduced the rate of albuminuria and the risk of DN. Another study analyzed renal outcomes in 19,664 diabetic and 11,648 non-diabetic patients over a 56-month follow-up period [6]. Based on the patients’ self-reports of their physical activity levels, individuals who engaged in physical activity 2–6 times per week had a 43% lower relative risk of adverse kidney outcomes compared to sedentary individuals, while individuals who exercised daily had a 44% lower relative risk. In the same vein, a cohort study validated that increased cardiorespiratory fitness was linked to a reduced predictive risk of mortality among individuals with CKD [11]. Studies by Hawley et al. [12] showed that exercise training can improve insulin sensitivity in insulin-resistant individuals, possibly due to the enhanced expression and activity of proteins related to glucose metabolism, insulin transduction, and enhanced lipid oxidation in muscle. Participating in exercise has the potential to enhance cardiorespiratory fitness. A randomized controlled trial conducted among 99 individuals with a mean GFR of 33 mL/min/1.73 m^2^, where 59% had diabetes and 29% had coronary artery disease, demonstrated that long-term exercise training leads to an improvement in cardiorespiratory fitness among patients with chronic kidney disease (CKD). This improvement is accompanied by an enhanced GFR and a reduction in albuminuria [13]. Patients with DN have a higher mortality rate than CKD patients without diabetes. In a cohort study of 1,024,977 patients with CKD (128,505 with diabetes), participants with diabetes had mortality risks 1.2 to 1.9 times higher compared to those without diabetes [14]. It seems that the positive effects of physical exercise on kidney health can be especially potent for individuals with diabetes [6]. 

The progression of DN has shown an inverse relationship with both the intensity and frequency of exercise. A 10-year follow-up investigation involving 2180 individuals with type 1 diabetes revealed that engaging in exercise, especially at a high frequency and intensity, was associated with a decreased risk of DN advancement [15]. In longitudinal studies of alumni from the University of Pennsylvania and Harvard University, researchers followed them up to investigate the impact of physical activity on hypertension risk, classifying physical and recreational activities as light, mixed, and vigorous based on calories burned per minute (5, 7.5, and 10 kilocalories per minute). They noted that the risk of hypertension was inversely correlated with vigorous sports participation [16]. Preventing hypertension is known to improve kidney function, thereby decreasing the death rate. In a prospective study with a 6.4-year follow-up of 1390 patients with type 1 diabetes of at least 20 years in duration [17], the author focused on the renal outcomes in these patients. Based on patient-self-reported exercise levels, they found that, in terms of exercise intensity, patients who exercised at low, medium, and high intensities had 10-year cumulative mean progression rates of kidney injury of 24.0%, 13.5%, and 13.1%, respectively. In terms of exercise frequency, patients who exercised at low, moderate, and high frequencies had kidney injury progression rates of 24.7%, 14.7%, and 12.6%, which suggests that a higher intensity and higher frequency of exercise may slow the progression of DN more effectively. Pongrac Barlovic et al. [18] confirmed that engaging in regular moderate to vigorous exercise is linked to a decreased likelihood of both the onset and advancement of kidney disease. Furthermore, it is also associated with a reduced risk of cardiovascular events and mortality.

Several studies have explored the ways in which exercise may play a protective role in the kidneys. The study of Baião et al. [19] showed that long-term exercise intervention significantly reduced the levels of C-reactive protein (CRP) (SMD: −0.23; 95% CI: −0.39 to −0.06) and pro-inflammatory cytokines such as IL-6 (SMD: −0.35; 95% CI: −0.57, −0.14) and TNF-α (SMD: −0.63, 95% CI: −1.01, −0.25) in patients with CKD, and increased the level of IL-10 (SMD: 0.66, 95% CI: 0.09, 1.23), proving that exercise intervention can alleviate kidney damage by relieving inflammation. The level of IL-6 released by human muscles after exercise is positively correlated with exercise intensity [20]. IL-6 is often seen as a pro-inflammatory cytokine, however, IL-6 produced after exercise appears to induce the production of an anti-inflammatory environment [21,22]. Intense exercise stimulates the production of IL-6, which subsequently elevates the plasma concentrations of two anti-inflammatory cytokines, namely IL-1 receptor agonists (IL-1ra) and IL-10 [23]. In addition, the increase in IL-6 levels after exercise is only a temporary response, and the result of long-term exercise intervention is a decrease in plasma IL-6 levels [24]. Costanti-Nascimento et al. [25] showed that moderate exercise can regulate the immune system to develop in an anti-inflammatory direction in diabetic kidney injury, and exercise can reduce the expression of inflammatory factors and TGF-β and alleviate oxidative stress to in order alleviate diabetic kidney injury. Therefore, exercise may reduce inflammatory and fibrotic factors, as well as alleviate oxidative stress in the kidneys to play a protective role.

Although the best types of exercise, frequency, intensity, and duration to slow the progression of DN are still inconclusive, several studies have made some suggestions. The study of Wilkinson et al. [26] pointed out that exercise that can continuously mobilize the major muscle groups of the whole body is the most effective in alleviating CKD, such as walking and a combination of resistance exercise and aerobic exercise, which can improve the efficiency of aerobic exercise by improving muscle mass. Karstoft et al. [27] divided diabetic patients into two groups: intermittent walking training and continuous walking training; the continuous walkers maintained a walking pace at 55% of their peak energy expenditure rate, whereas the interval walkers alternated between fast walking (above 70% of their peak energy expenditure) and slow walking (below 70% of their peak energy expenditure) for 3-min intervals. All participants consistently practiced these activities five times a week in 60-min sessions for 4 months, and it was discovered that interval walking outperformed continuous walking matched for energy expenditure in enhancing physical fitness, body composition, and glycemic control. As for water-based exercise, Pechter et al. [28] showed that all CKD patients who participated in water-based exercise showed a significant reduction in proteinuria and serum cystatin C compared to the sedentary group, confirming an improvement in kidney function. Pechter et al. [29] also investigated the possible beneficial renal effects of regular water-based exercise in patients with CKD. After 12 weeks of swimming exercise for 30 mins twice a week, it was shown that the above exercise program improved all cardiopulmonary function parameters, significantly reduced proteinuria, and increased the GFR in the patients. Interestingly, water-based exercise is more beneficial for the improvement in kidney function than land exercise. In a 6-month exercise program with female spontaneously hypertensive rats swimming for 30 min a day, three days a week, water immersion provided an ideal environment to relieve the excessive activity of the sympathetic nerves in the kidney [30]. In addition, the increased constriction of the kidney blood vessels due to the upright posture of land exercise can be avoided during water exercise, which indicates that water-based exercise may improve DN more effectively than land exercise [29]. 

In summary, mounting evidence indicates that exercise can alleviate the progression of renal damage in patients with DN (Figure 1A). In addition, the intensity, frequency, and type of exercise are also important factors. It is recommended that DN patients should engage in exercise that mobilizes the whole body’s major muscle groups, considering a combination of land exercise, water exercise, and resistance training. More appropriate exercise programs for DN patients need to be further studied in the future.

## 3. Amelioration of DN by Exercise in Animal Studies

The role of exercise in ameliorating DN has also been extensively studied by using animal models. Elevated oxidative stress in diabetes causes damage to the kidneys. Glutathione is important for maintaining redox homeostasis in many organs, including the kidneys. Afolayan and Sunmonu’s study showed that diabetic rats saw a considerable decrease in the expressions of glutathione peroxidase and glutathione reductase, two important antioxidant enzymes. However, moderate exercise resulted in an increase in the levels of these antioxidant enzymes [31,32]. Yamakoshi et al. [33] showed that 12 weeks of exercise training (20 m/min, 60 min/day, 5 days/week) in rats with chronic kidney disease reduced the expressions of NADPH oxidase and xanthine oxidase in the kidneys, thereby reducing the oxidative stress of the kidneys. Inflammation is another important factor that causes kidney damage, and markers of inflammation are significantly higher in patients with CKD than in healthy people [34]. Souza et al. [35] investigated the effect of exercise on DN in ovariectomized rats with streptozotocin (STZ)-induced diabetes; the exercise protocol consisted of four weeks of progressive exercise 60 min per day, five days per week, for eight weeks at 70% of their maximal exercise capacity. They found that the number of ED-1- and CD43-positive cells were increased, which indicates that the accumulation of macrophages and lymphocytes and the immunostaining of nuclear factor kappa-B (NF-κB) were increased in the glomeruli and tubules of the diabetic rats. Nonetheless, physical exercise resulted in a reduction in the infiltration of immune cells and the downregulation of NF-κB expression in the kidneys of the diabetic rats. Consistently, Ishikawa et al. [36] found that exercise reduced macrophage infiltration, T lymphocyte accumulation, and NF-κB pathway activation in the kidneys of type 2 diabetic mice.

There is an interaction between inflammation and oxidative stress. When free radical/reactive oxygen species (ROS)-mediated nephron damage occurs, inflammation subsequently appears and stimulates additional free radical formation. Neutrophils, acting as phagocytes alongside other phagocytic cells, are mobilized to the injured nephron to generate superoxide through their membrane-associated nicotinamide adenine dinucleotide phosphate (NADPH) oxidase system [37,38]. Superoxide and other free radicals, along with their modified targets, persist in either worsening kidney injury or acting as signaling molecules that initiate a sustained inflammatory response within the kidney. This additional pro-inflammatory signal inevitably leads to the formation of more free radicals/ROS and sustained damage to nephrons [39,40]. Excessive ROS production within the kidneys, along with the accumulation of protein products containing dityrosine (referred to as advanced oxidation protein products, AOPPs) generated during oxidative stress, have been directly associated with podocyte injury, proteinuria, and the development of conditions such as focal segmental glomerulosclerosis (FSGS) and tubulointerstitial fibrosis [41].

NADPH oxidase (NOX) family enzymes work to produce ROS that ultimately cause inflammatory stress. NOX4, which is extensively expressed in the kidneys, is an important source of ROS in the kidneys [42]. Cepas et al. [43] reported that AGEs up-regulate NOX enzymes’ expression in a receptor for AGEs in a (RAGE)-dependent manner to mediate an increase in intracellular ROS, thus increasing oxidative stress. AGEs that build up on the walls of blood vessels cause oxidative stress and inflammation, which damages endothelial cells in the glomeruli [44], leading to the progression of DN. Exercise can reduce AGEs. Boor et al. [45] investigated whether regular moderate exercise in obese Zucker rats (OZR), a model of diabetes and obesity-associated nephropathy, influenced the development of kidney injury by reducing the formation of AGEs. The exercise program for the exercise group consisted of a 5-week conditioning phase with a gradual increase in exercise intensity up to the ten weeks of continuous training, and the results showed that the AGEs in the kidneys of DN rats decreased significantly after exercise, which may have been related to the fact that the high energy requirements induced by exercise may reduce the pool of active intermediates (glycolysis intermediates and polyols) available for glycation [46].

Kotake et al. [8] randomly divided male Torii rats with spontaneous diabetes into a sedentary and exercise group. The exercise group adopted a gradual exercise program with a 3° slope on treadmill, where the speed increased from 10 m/min to 20 m/min, for 30 min a day, four times a week. The albuminuria levels were significantly lower in the exercise group than in the sedentary group, and glomerular endothelial cell damage and oxidative stress levels were also reduced in the exercise group. In addition, the exercise group had reduced diabetes-induced glomerular area enlargement, less damage to the superficial and intermediate layers of the renal cortex, and less tubule interstitial fibrosis. Consistently, Amaral et al. [47] performed exercise interventions in diabetic rats. They found that exercise alleviated proteinuria, glomerular hypertrophy, and tubular and glomerular damage, and decreased the expressions of fibronectin and collagen IV related to renal fibrosis in the kidneys of diabetic rats.

Taken together, exercise is able to ameliorate the progression of DN in animal models of DN. Various studies have shown that exercise reduces macrophage infiltration in the kidneys, the activation of inflammatory signaling pathways, and the production of pro-inflammatory cytokines, and decreases the accumulation of AGEs and ROS in the kidneys. In the ways above, exercise plays a protective role against DN in animal models, which can alleviate injury to podocytes and endothelial cells, reduce albuminuria, and delay the development of glomerulosclerosis and tubulointerstitial fibrosis (Figure 1B).

## 4. Molecular Mechanism of Exercise-Mediated Alleviation of DN

### 4.1. Role of Exercise-Mediated miR-181b Up-Regulation in Amelioration of DN

MicroRNAs, often abbreviated as miRNAs, are a group of naturally occurring, short, single-stranded RNAs, typically measuring around 20–23 nucleotides in length, which play a role in the regulation of gene expression [48]. In DN, multiple miRNAs appear to be aberrantly regulated, which promotes inflammation and fibrosis [49]. In members of the miR-181 family (miR-181a, miR-181b, miR-181c, and miR-181d), miR-181b exhibited a negative correlation with vascular inflammation in individuals diagnosed with type 2 diabetes [50]. Ishii et al. compared the differences in miRNAs’ expression in the kidney tissue of DN mice with those of control mice. Among 125 differentially expressed miRNAs, only miR-181b was downregulated, and the treatment of DN mice with miR-181b reduced albuminuria and abnormal mesangial dilatation in their kidney tissues [51]. Cheng et al. [52] found that the miR-181b levels in the blood flow were reduced in diabetic patients and could be increased by exercise. MiR-181b can attenuate vascular inflammation and endothelial dysfunction in diabetic mice. In diabetic conditions, AGEs can accumulate on the walls of blood vessels and damage endothelial cells by causing oxidative stress and inflammatory responses that disturb the surface of the endothelial cells. The incubation of AGEs resulted in the downregulation of miR-181b in endothelial cells [44].

MiR-181b is a downstream target of 5-AMP-activated protein kinase (AMPK) signaling. Aerobic exercise may upregulate the expression of fibronectin type III domain 5 protein (FNDC5)/Irisin through the increase in muscle blood flow during exercise, and this increase in exercise-activated irisin expression may bind to the αV-integrin receptor on kidney cells and activate AMPK in the kidneys somehow [53]. Han et al. [54] confirmed that the activation of renal AMPK can promote the transfer of Pink1 from the cytoplasm to the mitochondria, thus promoting mitochondrial autophagy. Witkowski et al. [50] showed that miR-181b directed its action towards phosphatase and tensin homolog (PTEN), resulting in a decrease in PTEN expression, thereby reducing endothelial cell inflammation. The lipid phosphatase domain of PTEN catalyzes the dephosphorylation of phosphatidylinositol-3,4,5-bisphosphate (PIP3) at the 3′ position, converting it back into phosphatidylinositol-4,5-bisphosphate (PIP2) [55]. By reducing intracellular PIP3 levels, PTEN inhibits the activation of serine/threonine kinase AKT, a protein downstream of the Phosphoinositide 3-kinase (PI3K) pathway [56]. The activation AKT phosphorylates a wide range of downstream targets mediating a variety of cellular processes. For example, AKT inhibits the BAD signaling pathway, thus inhibiting cell apoptosis [57]. Furthermore, AKT inhibits apoptosis by phosphorylating forkhead transcription factors [57]. Past research has demonstrated that the activation of the PI3K-Akt pathway in human monocytes restricts the expression of TNF-α by decreasing the nuclear translocation of NF-κB [58]. Therefore, exercise could increase miR-181b to blunt the PTEN-mediated inhibition of PI3K/AKT signaling, thereby suppressing vascular inflammation, cell apoptosis, and kidney injury in DN (Figure 2).

### 4.2. Role of Exercise-Regulated Renin–Angiotensin System in Amelioration of DN

The onset of diabetes results in the excessive activation of the renal renin–angiotensin system (RAS), which, in turn, triggers a cascade of inflammation and fibrosis within the kidneys, ultimately causing a decline in kidney function [59]. The RAS is an important system in the body. Renin (or angiotensinogenase) is secreted by the kidney granular cells. It hydrolyzes angiotensinogen secreted by the liver into angiotensin I (Ang I). Angiotensin-converting enzyme (ACE) converts Ang I into angiotensin II (Ang II). Ang II is a growth factor which has the effect of constricting the blood vessels, and its overactivation can cause inflammation. ACE2 has antagonistic effects against ACE, which can degrade Ang II to produce the vasodilatory heptapeptide Ang-(1-7), an essential factor for maintaining renal function homeostasis. A deficiency in ACE2 can lead to albuminuria and glomerular injury [60] (Figure 3A).

A study performed resistance exercise training on DN rats and indicated that the activities of both ACE and ACE2 were significantly modulated following resistance exercise. Decreased ACE activity and increased ACE2 activity were observed in the kidneys of the DN rats undergoing resistance training, leading to decreased levels of inflammatory cytokines. Resistance exercise reduced the levels of Ang I and Ang II and increased the concentration of Ang-(1-7) in the kidneys [61]. Ang II can promote kidney inflammation in DN rats, while Ang-(1-7) can counteract the effect of Ang II on promoting kidney inflammation. An increase in Ang-(1-7) can enhance the anti-inflammatory process, which has a protective effect on the kidneys [62]. The elevated levels of Ang-(1-7) detected in the kidneys of DN rats undergoing resistance training could be closely related to the enhanced ACE2 activity. The actions of Ang II and Ang-(1-7) are modulated through their engagement with distinct plasma membrane receptors: Ang-II type 1 receptor (AT1R) for Ang II and Mas receptor (MasR) for Ang-(1-7), both belonging to the family of G-protein-coupled receptors. Ang II stimulates AT1R, resulting in increased blood pressure (BP) and vascular wall fibrosis. Conversely, Ang-(1-7) activates MasR, eliciting effects that contrast with Ang II [63]. ACE2 enzyme-mediated degradation of Ang II is the basis of the treatment of DN [64]. Therefore, the overactivation of RAS in diabetes can aggravate DN. However, exercise can boost the activity of renal ACE2 and reduce the activity of renal ACE, which will promote the shift of renal RAS towards the ACE2/Ang-(1-7) axis, reduce renal inflammatory cytokine levels, and protect against DN (Figure 3B).

### 4.3. Role of Exercise-Mediated Increase in Sirt1 in Amelioration of DN

Sirtuins, classified as class III histone deacetylases, have significant roles in regulating multiple biological functions, including apoptosis, DNA repair, cell cycle, mitochondrial function, oxidative stress, energy metabolism, and aging [65]. Sirtuin 1 (Sirt1) can be expressed in the kidney podocytes, mainly located in the nucleus, and exerts its effects to reduce cellular senescence [66]. Chuang et al. [67] investigated the effect of Sirt1 in adriamycin-induced nephropathy, and observed that those with kidney-specific Sirt1 knockdown exhibited more pronounced albuminuria and mitochondrial dysfunction when compared to diabetic mice without renal Sirt1 knockdown. Moreover, the overexpression of Sirt1 in human proximaltubular epithelial cell line (HK-2) treated with high glucose reduced the activation of NF-κB [68]. In addition, Huang et al. [69] found that Sirt1 overexpression prevents the production of ROS in glomerular mesangial cells (GMCs) challenged with AGEs. Therefore, Sirt1 has potential in delaying the progression of DN.

Kidney diseases may be associated with a poor mitochondrial status. Mitochondrial function can be regulated by some key factors, including AMPK, sirtuins, and peroxisome proliferator-activated receptor γ coactivator-1alpha (PGC1-α), which play important roles in the fight against kidney disease [70]. Sirt1 stimulates mitochondrial biogenesis through the deacetylation and activation of PGC1-α. PGC1-α is highly expressed in the region with high cellular respiration in the kidney (the renal cortex and corticomedullary junction) [71]. Tang et al. [72] found that the expressions of Sirt1 and PGC1-α are reduced in the kidneys of diabetic mice, and aerobic exercise training partially reversed this change; meanwhile, mitochondrial superoxide production was significantly reduced and mitochondrial membrane potential and ATP production were increased, indicating that mitochondrial function was improved by exercise. The mechanism may be that the expression of Sirt1 is increased by exercise training, which promotes the activity and expression of PGC1-α.

Yang et al. [73] found that exercise resulted in an increased expression of renal Sirt1 in diabetic mice, concomitant with the suppression of the p53-mediated pro-apoptotic pathway, which is related to reduced tissue fibrosis and an increased GFR. In vitro experiments have shown that H2S treatment can restore Sirt1 expression in high-glucose-treated glomerular podocytes (MPC5 cells). Cystathionine-β-synthase (CBS) and cystathionine-γ-lyase (CSE) are enzymes responsible for facilitating the synthesis of endogenous H2S and are predominantly expressed in the kidneys, their study also showed that exercise increased the expressions of CBS and CSE, accompanied by increased H2S production in DN mice. These results indicate that exercise training enhanced the production of endogenous H2S in kidney tissues, which contributed to the increase in renal Sirt1 expression and the inhibition of p53-mediated cell apoptosis, thereby promoting improvement in DN.

Liu et al. [74] investigated the metabolic and inflammatory signaling in the kidneys of db/db mice with or without exercise training. They discovered that exercise averted the decrease in Sirt1 expression in DN mice, a phenomenon associated with decreased acetylation of NF-κB in the kidneys. Studies have reported that the upregulation of Sirt1 expression in the kidney is negatively correlated with NF-κB activity, and the downregulation of Sirt1 in the kidneys leads to p65 acetylation and subsequent NF-κB activation, leading to sustained inflammatory responses [75,76]. Sirt1 can inhibit NF-κB signaling either by deacetylating the p65 subunit of the NF-κB complex or by inhibiting ROS-mediated IκBα degradation [77]. Thus, decreased Sirt1 expression in DN is associated with increased NF-κB activation, in which increased NF-κB acetylation and enhanced ROS-mediated IκBα degradation are involved. As a result, exercise can inhibit NF-κB activity by restoring Sirt1 expression in the kidneys.

Therefore, in the kidneys of diabetic mice undergoing aerobic exercise training, the production of endogenous H2S could increase the expression of Sirt1, which leads to increased activity of PGC1-α and the inhibition of NF-κB activity, resulting in improved mitochondrial function and reduced inflammation in diabetic kidneys (Figure 4).

### 4.4. Role of Exercise-Mediated Increase in NO in Amelioration of DN

NO plays a key signaling role in the body. Adequate levels of NO reduce vascular resistance, thereby maintaining an adequate blood flow to maintain a healthy vascular system [78]. NO plays a role in modulating tubular transport in the kidney. The restoration of NO homeostasis is therapeutically valuable for the alleviation of renal disease [79]. Impaired function of the nitric oxide synthase (NOS) system or increased oxidative stress reduce nitric oxide (NO) bioavailability and lead to endothelial dysfunction [80]. NOS includes three isoforms: endothelial NOS (eNOS), neuronal NOS (nNOS), and inducible NOS (iNOS) [81]. Earlier investigations have documented the downregulation of endothelial nitric oxide synthase (eNOS) and neuronal nitric oxide synthase (nNOS) expressions in the kidneys of rats suffering from chronic renal failure [82]. Brodsky et al. [83] found that exposing human endothelial cells to elevated glucose levels leads to an impaired chemical-induced release of NO. Rodrigues et al. [84] trained DN rats for 16 m/min, 60 min/d, 5 d/week for eight weeks; urinary NO was lower in the DN group compared to the control group, possibly because of the NO-scavenging properties of high glucose, directly or through increased superoxide anions [85]. The exercised DN group showed an increase in urinary NO concentration compared to the sedentary DN group. The elevated NO levels observed in the exercised DN group may signify improved glycemic control and a diminished level of oxidative stress.

To investigate how exercise affects DN progression through NO, Ito et al. [86] trained diabetic obese rats at a speed of 20 m/min for 60 min/d, 5 days/week for 8 weeks. In comparison to the sedentary group, exercise resulted in decreased urinary albumin excretion, improved creatinine clearance, and mitigated kidney injury. Additionally, exercise upregulated the expressions of endothelial nitric oxide synthase (eNOS) and neuronal nitric oxide synthase (nNOS) in the kidneys while reducing NADPH oxidase activity, as well as the expressions of p47phox and α-oxoaldehydes, which are the precursors for AGEs. NADPH oxidase is a multi-subunit enzyme that includes prominently expressed p47phox in podocytes [87]. The renal cortex of rats with DN exhibited increased activity of NADPH oxidase and expression of the p47phox subunit protein. This indicates that NADPH oxidase may play a role in the elevation of superoxide (O_2_^−^) levels in rat kidneys, facilitated by the involvement of the p47phox subunit. The rapid conversion of O_2_^−^ into peroxynitrite (ONOO^−^) reduces the bioavailability of NO [88]. Another possible mechanism of the exercise-mediated increase in renal NO bioavailability could be the occasional change in shear stress on the vascular system mediated by exercise. Tinken et al. [89] used cuff dilation to control both anterograde and retrograde blood flow and shear force during hand grip exercise for 8 weeks. They found that an exercise-induced increase in vascular shear force improved endothelial function and demonstrated an increase in NO with an increase in shear force during hand grip training. Thus, exercise-induced changes in shear stress in blood flow could underlie the mechanism of exercise-mediated increases in NO levels in rat kidneys [90]. Therefore, by controlling the blood glucose, increasing the expressions of eNOS and nNOS, decreasing oxidative stress, and changing shear stress in the blood flow, exercise can elevate the renal NO level, which plays a protective role against DN.

### 4.5. Role of Exercise-Mediated Inhibition of P2X7 Receptors in Amelioration of DN

In the presence of inflammation, ATP accumulates in damaged and inflamed tissues. Its effects are mediated by plasma membrane receptors known as P2 receptors (P2Rs). Among these receptors, the P2X7 receptor (P2X7R) is particularly linked to inflammation and immunity [91]. Solini et al. [92] investigated the association of diabetes mellitus type 2 (T2DM) or metabolic-syndrome-related kidney disease with P2X7R, and found that mice lacking P2X7R had reduced inflammation, fibrosis, and oxidative stress in the kidney with a downregulation of NLRP3 inflammasome activation, suggesting that P2X7R and NLRP3 could be used as therapeutic targets for DN. Rodrigues et al. [93] evaluated the changes in P2X7R expression and activity in the kidneys of diabetic rats undergoing aerobic training; they trained DN rats on a treadmill, and after 8 weeks of 60 min/d, 16 m/min, 5 d/week treadmill exercise without inclination, they found that exercise led to a decrease in the expression and activity of P2X7R in the kidney tissues of the diabetic rats. This was accompanied by the control of oxidative stress, the protection of glomerular structures, and reductions in proteinuria and renal injury.

The contribution of P2X7R to worsening DN might be linked to elevated oxidative stress. The interplay of oxidative stress and high blood sugar triggers the activation of the p38 mitogen-activated protein kinase (p38MAPK) signaling pathway, leading to heightened levels of extracellular ATP and an upregulation in the expression of P2X7R [94,95]. As injury progresses, more P2X7R receptors are expressed in the renal tubular epithelium. Yi Zhou et al. confirmed that extracellular ATP acted by activating P2X7R gated channels [96], which activated nucleotide-binding oligomeric domain-like receptor protein 3 (NLRP3) inflammasome in renal tubular epithelial cells, and blockade of the P2X7R receptor axis protected renal tubular epithelial cells by downregulating the activation of the NLRP3 inflammasome [97]. This indicates that extracellular ATP may bind and activate P2X7R on the cell membrane, thereby leading to NLRP3 activation. At the same time, the P2X7R receptor can also be activated by intracellular ROS. ROS function as agonists by binding to the intracellular terminals of receptors, resulting in the activation of free radicals and voltage-dependent Ca^2+^ channels. The activation of P2X7R increases cell membrane permeability and Ca^2+^ inward flow, which may cause cell death [93]. In addition, a study by Bourzac et al. [98] reported that P2X7R activation downregulated the activity of glucose transporter-2 on the plasma membrane of kidney cells, which can reduce the absorption of glucose and promote uncontrolled high blood sugar. Therefore, exercise can reduce the activation and expression of P2X7R in the renal tubular epithelium to blunt the ATP-mediated activation of NLRP3 inflammasomes, reduce cell death caused by Ca^2+^ influx, and reduce hyperglycemia by upregulating glucose transporter-2, thereby alleviating DN (Figure 5).

### 4.6. Role of Exercise-Mediated Increase in Heat Shock Protein in Amelioration of DN

HSPs degrade misfolded proteins to prevent their accumulation, thus achieving a protective effect on cells [99]. HSP72, a member of the HSP70 family, is expressed at a high level in the inner medulla of the kidney and plays an important role in protecting cells from high urea toxicity [100]. HSP72 can be widely activated in a variety of tissues in response to exercise [101]. HSP72 has demonstrated the capacity to enhance oxidative metabolism and insulin sensitivity, a critical mechanism in the management of type 2 diabetes [102]. Due to the changes in the blood glucose and osmotic pressure in the kidneys during diabetes, diabetic kidneys have a higher demand for energy metabolism, which increases the demand for HSPs [103]. In an animal model of chronic renal tubulointerstitial fibrosis, the induction of HSP72 expression attenuated renal tubular cell apoptosis and interstitial fibrosis [104]. Lappalainen et al. [103] investigated the effects of exercise on HSR in the kidneys of diabetic rats, and their study showed that exercise training substantially increased the levels of kidney HSP72 and notably reduced oxidative stress in diabetic kidneys. An exercise-induced increase in HSP72 expression was associated with decreased levels of TGF-β, IL-6, and TNF-α in the kidneys. In addition, an improvement in glomerular sclerosis was also observed in the exercise group. Therefore, exercise can promote the expression of HSP72, which contributes to the alleviation of glomerular injury and DN. The mechanism by which exercise-induced HSP72 ameliorates kidney injury needs further study.

### 4.7. Role of Exercise-Mediated Gut Microbiota Changes in Amelioration of DN

CKD can be affected by changes in gut microbes, according to many studies. The composition and expression of gut microbes varied widely between cohorts of patients with CKD and controls of healthy people [105]. In one case, CKD was associated with a loss of resident microbial flora, while in another, gut dysbiosis affected the progression of CKD [106]. Yao et al. [107] evaluated the microbial signatures associated with the development of diabetes in a cohort of student athletes, and found that 12 exercise-induced changes in gut bacteria were associated with a reduced risk of diabetes. Probiotics that ameliorate intestinal disorders such as Lactobacillus caseii have been shown to improve kidney inflammation, possibly by increasing the level of short-chain fatty acids (SCFAs), which act on inflammatory cells and SCFA transporters such as Slc5a8 and Slc18a1 in proximal renal tubular epithelial cells [108]. Liu et al. [109] collected the feces of 39 diabetic patients after a 12-week exercise intervention and sequenced them. A Kyoto Encyclopedia of Genes and Genomes (KEGG) analysis was conducted to show that 214 intestinal microbial genes were significantly upregulated after exercise, and these changes were closely related to the enhancement of insulin sensitivity. Moreover, the metabolic pathway for the biosynthesis of SCFAs was increased, and an increase in SCFAs was also observed in the exercise group. Therefore, this exercise-induced alteration of the gut microbiome may alleviate kidney inflammation in DN by increasing SCFAs’ production, which needs further study.

Wertheim et al. [110] analyzed the relationship between fecal bile acid levels and exercise in 735 patients with colorectal adenomas, and found that an increase in the duration of physical activity was associated with a decrease in fecal bile acid concentrations, possibly because of an exercise-mediated improvement in the bile acid absorption efficiency in the intestines. Receptors of bile acid are nuclear hormone receptor farnesol X receptor (FXR) and G protein-coupled receptor (TGR5), which are highly expressed in the kidneys. Wang et al. [111] treated diabetic renal injury mice with an FXR/TGR5 dual agonist, and they found that proteinuria, podocyte injury, mesangial dilation, and renal tubulointerstitial fibrosis of the mice were alleviated after the agonist treatment. Thus, exercise may alleviate kidney injury in DN by improving the efficiency of bile acid absorption to better activate FXR/TGR5 signaling in the kidney, which needs further investigation. Moreover, changes in the metabolites of gut microbes also play an important role in the progression of kidney disease. For example, choline, phosphatidylcholine, and L-carnitine in the intestine are metabolized to trimethylamine (TMA) by intestinal symbiotic bacteria, and TMA is oxidized to trimethylamine-n-oxide (TMAO) in the liver. An increase in TMAO in the circulating blood causes kidney damage by promoting inflammation and oxidative stress in the kidneys [112]. In a 12-month follow-up of 483 people with diabetes, Argyridou et al. [113] found that moderate to vigorous exercise was associated with a reduction in TMAO. Erickson et al. [114] conducted a combined exercise and low-calorie intervention in 16 obese adults for 12 weeks, and found that a low-calorie diet and exercise intervention could effectively reduce plasma TMAO levels. Therefore, it is reasonable to speculate that exercise can alleviate kidney injury in DN by regulating the homeostasis of the gut microbiota to reduce TMAO levels. More studies are needed to verify this association and the mechanisms involved (Figure 6).

### 4.8. Role of Exercise-Induced Hormones and Metabolites in Amelioration of DN

#### 4.8.1. Neuregulin 4 (Nrg4)

Nrg4 is an adipokine whose circulating expression levels rise after exercise [115]. In a study involving 140 diabetic patients, decreased serum Nrg4 levels were strongly associated with the development of DN compared with controls [116]. In a cross-sectional study, serum Nrg4 expression was significantly reduced in 60 ESKD patients compared to controls. In addition, in mouse models with DN, reduced Nrg4 expression levels were also observed in adipose tissue, which was independently associated with an impairment of kidney function [117]. Shi et al. [118] found that an Nrg4 treatment of DN rats can reduce the accumulation of AGEs, downregulate inflammatory cytokines, and improve renal tubule interstitial fibrosis, thus alleviating DN. They treated HK-2 cells with TNF receptor 1 (TNF-R1) and found that TNF-R1 overexpression inhibited the remission of DN by Nrg4, which indicates that Nrg4 may delay DN progression by inhibiting TNF-R1. Therefore, the elevation of Nrg4 may be involved in the exercise-mediated alleviation of DN, which is worthy of further study (Figure 7).

#### 4.8.2. Irisin

Irisin is the extracellular moiety of FNDC5. Endurance training can induce the release of irisin from muscle or adipose tissue, possibly due to the activation of its upstream regulator peroxisome proliferator-activated receptor γ coactivator-1α (PGC-1α) by exercise [119,120]. Formigari et al. [53] discovered that FNDC5/Irisin and its upstream protein PGC-1α were upregulated by aerobic exercise in the muscles of diabetic rats, which significantly correlates with the decreases in albuminuria and glomerular fibronectin expression and the acetylation of NF-κB. By treating diabetic rats and HK-2 (renal tubular epithelial cells) with an irisin receptor inhibitor (the αV-integrin receptor blocker) and recombinant irisin, it was shown that irisin-mediated signaling was related to the activation of AMPK. However, the mechanism through which exercise-induced irisin activates AMPK in the kidneys remains unclear, and it is speculated that irisin may bind to the αV-integrin receptor on the kidney cells, causing renal AMPK activation somehow. In addition, the study of Lai et al. [121] showed that increased irisin in plasma can inhibit the overactivation of the high-glucose-induced PI3K/AKT/mTOR signaling pathway in human podocytes, thereby restoring podocyte autophagy. Therefore, exercise may ameliorate kidney injury by activating the AMPK signaling pathway or inhibiting theAKT signaling pathway through irisin secreted by the skeletal muscles during exercise, which needs to be further investigated (Figure 7).

#### 4.8.3. Metrnl

Metrnl is a novel secreted protein which is homologous to meteorin (Metrn) and can be induced in the skeletal muscles after exercise [122]. The expression and function of Metrnl have been widely discussed. Metrnl is recognized for its significant involvement in white adipose tissue metabolism, enhancing the functionality of fat cells and addressing insulin resistance associated with obesity [123]. In order to determine the potential role of Metrnl in the progression of DN, Wang et al. [124] conducted a study in which 221 T2DM patients and 74 healthy controls were included. Patients with T2DM were divided into three groups: normal albuminuria, microalbuminuria, and macroalbuminuria groups, according to the UACR. It was found that, compared with the other T2DM subgroups, the serum Metrnl concentration was significantly lower in T2DM with macroalbuminuria, serum Metrnl was negatively correlated with the course of DN, blood urea nitrogen, creatinine, uric acid, and UACR, which indicated that serum Metrnl concentration was negatively correlated with the severity of DN. During exercise, the PGC-1α4 protein level is increased in exercising muscles, and PGC-1α4 stimulates Metrnl mRNA expression, which leads to the increased release of Metrnl protein into the bloodstream from the skeletal muscles [125]. Jung et al. [122] found that Metrnl released by exercise significantly increased the expression of PPARδ and increased the phosphorylation of AMPK in the skeletal muscles of mice, thus weakening the NFκB-mediated signaling pathway, decreasing the expression levels of pro-inflammatory cytokines (TNF-α and MCP-1), and alleviating insulin resistance and inflammation. It is possible that the Metrnl/AMPK pathway may be involved in the exercise-mediated alleviation of DN, which warrants further investigation (Figure 7).

#### 4.8.4. β-Hydroxybutyrate (BHB)

BHB is synthesized from fatty acids in the liver and can be released from the liver into other tissues after exercise, and it plays a role as a signaling metabolite in a variety of diseases [126]. Regular exercise training for 12 weeks in mice significantly upregulated their serum BHB levels and inhibited the progression of atherosclerosis [127]. A cohort analysis of 521 patients with Autosomal Dominant Polycystic Kidney Disease (ADPKD) showed that an increasing plasma BHB concentration is accompanied by an increase in GFR, and, therefore, increasing the BHB concentration in the plasma may mitigate the decline in kidney function [128]. Many studies have explored the protective effects of exogenous BHB supplementation on the kidneys. In AKI, treatment with BHB can alleviate the activation of the NLRP3 inflammasome and oxidative stress in the kidney, reducing the apoptosis of renal tubule cells [129]. Another study further investigated the mechanism by which BHB exerts a protective effect on the kidneys. Fang et al. [130] simulated a hyperglycemic environment by treating mouse podocytes with high sugar, and found that BHB alleviated the cell damage and senescence in podocytes induced by high sugar, possibly because of the BHB-mediated inhibition of glycogen synthase kinase 3β (GSK3β) activity, since GSK3β can catalyze the phosphorylation of transcription factor Nrf2 to promote its exit from the nucleus and a decrease in GSK3β activity promotes the accumulation of Nrf2 in the nucleus to play an antioxidant role. As a result, exercise may reduce the damage to podocytes by increasing the serum BHB level to inhibit the activity of GSK3β in podocytes, thus delaying the progression of DN, which deserves further investigation (Figure 7).

#### 4.8.5. β-Aminoisobutyric Acid (BAIBA)

BAIBA is a metabolite of valine that can be produced and secreted by exercising the skeletal muscles, acting on other tissues to exert its effects [131]. BAIBA plays an important role in mitigating metabolic diseases such as diabetes. Treating 3T3-L1 cells with BAIBA can reduce the production of pro-inflammatory cytokines and alleviate insulin resistance [132]. Some studies have proved that the exogenous treatment of BAIBA has a protective effect on kidney cells. Studies by Audzeyenka et al. [133] showed that the exogenous supplementation of BAIBA in podocytes can improve their mitochondrial respiration efficiency, which may be related to the activation of PGC-1α and transcription factor A mitochondria (TFAM) to enhance mitochondrial biogenesis in podocytes. In another study, NRK-49F (normal rat kidney interstitial fibroblast cells) was treated with BAIBA, and BAIBA could significantly inhibit the proliferation and migration of NRK-49F. This may have been caused by the block of the Ang II action and the decrease in oxidative stress induced by BAIBA [134]. Therefore, it is possible that BAIBA is also involved in the amelioration of DN by exercise, which awaits further investigation (Figure 7).

#### 4.8.6. Glucagon-like Peptide 1 (GLP-1)

GLP-1 is secreted from the gut endocrine L-cells, and it can reduce glucose concentration by inhibiting glucagon release and increasing insulin secretion [135]. Wu et al. [136] found that two weeks of exercise training in mice can promote the expression levels of GLP-1 in the gut and serum, and exercise can change the characteristics of the intestinal microbiota, such as increasing the abundance of bifidobacterium and lactococcus, etc., which is related to the increase in GLP-1 expression and secretion. The treatment of diabetic patients with GLP-1 receptor (GLP-1R) agonists can prevent the occurrence of large amounts of albuminuria, reduce the decline in the GFR, and play a protective role in the kidneys [137]. A meta-analysis showed that the treatment of T2DM with GLP-1R agonists resulted in a 17% reduction in adverse renal outcomes [138]. Winiarska et al. [139] showed that GLP-1 peptide therapy can significantly reduce the infiltration of macrophages and inflammatory cells in the kidney, and GLP-1 may ameliorate kidney injury through anti-inflammatory and antioxidant stress effects. Therefore, GLP-1 may also play a part in the exercise-mediated alleviation of DN, which merits further investigation (Figure 7).

## 5. Conclusions

In summary, DN is a common complication of diabetes and a leading cause of end-stage renal disease. In patients with DN, a decreased GFR, increased blood pressure, and higher cardiovascular risks are among the challenges they face. Exercise represents an effective strategy in alleviating the pathophysiological manifestations associated with DN. It exerts beneficial effects by delaying the decline in the GFR, reducing albuminuria, and attenuating inflammation and oxidative stress within the kidney. Furthermore, exercise plays a role in controlling the blood sugar and blood pressure, improving insulin sensitivity, and enhancing lipid metabolism, which are also beneficial for the improvement in kidney function. Animal studies have provided evidence for the protective effects of exercise on glomerular endothelial cells and podocytes, leading to a delay in the development of glomerulosclerosis and tubulointerstitial fibrosis. The beneficial effects of exercise on DN involve multiple molecular mechanisms. For instance, exercise activates Sirt1, promotes the activity and expression of PGC1-α, and improves mitochondrial function in the kidneys. Additionally, exercise regulates the renin–angiotensin system (RAS) to alleviate renal inflammation. Exercise also plays an anti-inflammatory role by regulating the gut microbiota. At the same time, some exerkines and metabolites released after exercise can be taken up in the kidney through the blood circulation to reduce kidney damage in DN. Although exercise has many beneficial effects against DN, the mechanisms through which exercise brings about these benefits need further and in-depth investigation.

In the future, more randomized controlled trials are needed to investigate the optimal intensity and frequency of exercise that can offer the best protection against DN, while avoiding exercise-related risks. It is also crucial to delve deeper into the mechanisms by which exercise alleviates DN and to promote the integration of exercise with nutritional and pharmacological interventions. Overall, exercise represents a highly promising strategy for the prevention and treatment of DN.

## Figures and Tables

**Figure 1 ijms-25-03605-f001:**
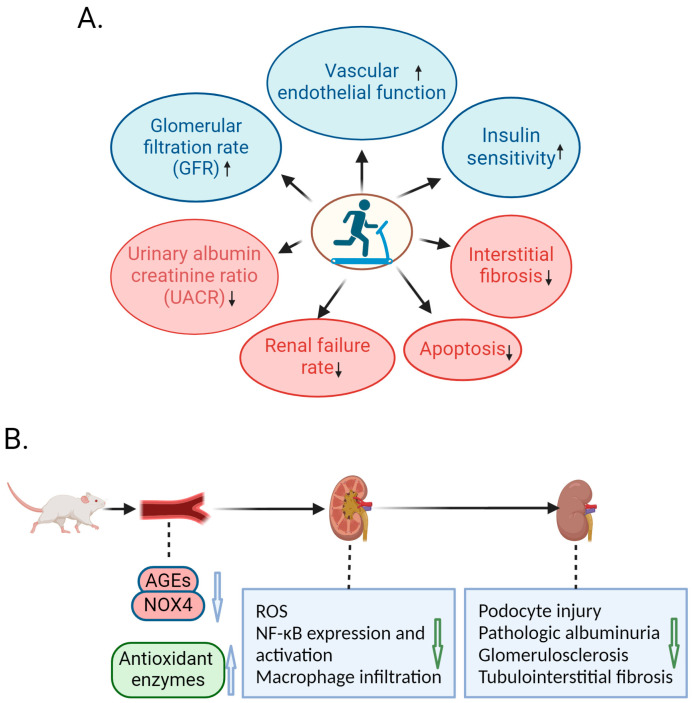
Protective effects of exercise against diabetic nephropathy (DN). (**A**) Human studies indicate that exercise can delay the progression of DN by increasing glomerular filtration rate (GFR), insulin sensitivity, and vascular endothelial function and decreasing urinary albumin creatinine ratio (UACR), interstitial fibrosis, apoptosis, and renal failure rate in patients with diabetes. The upward arrows inside the oval icons mean up-regulation and the downward arrows inside the oval icons mean down-regulation. (**B**) In DN animal models, exercise can decrease the expression levels of advanced glycation products (AGEs) and NOX4, increase the expression levels of antioxidant enzymes, and decrease ROS accumulation, NF-κB expression and activation, and macrophage infiltration, thereby alleviating podocyte injury, pathologic albuminuria, glomerulosclerosis, and tubulointerstitial fibrosis. The upward arrows mean up-regulation and the downward arrows mean down-regulation. Created with BioRender.com, accessed on 12 March 2024.

**Figure 2 ijms-25-03605-f002:**
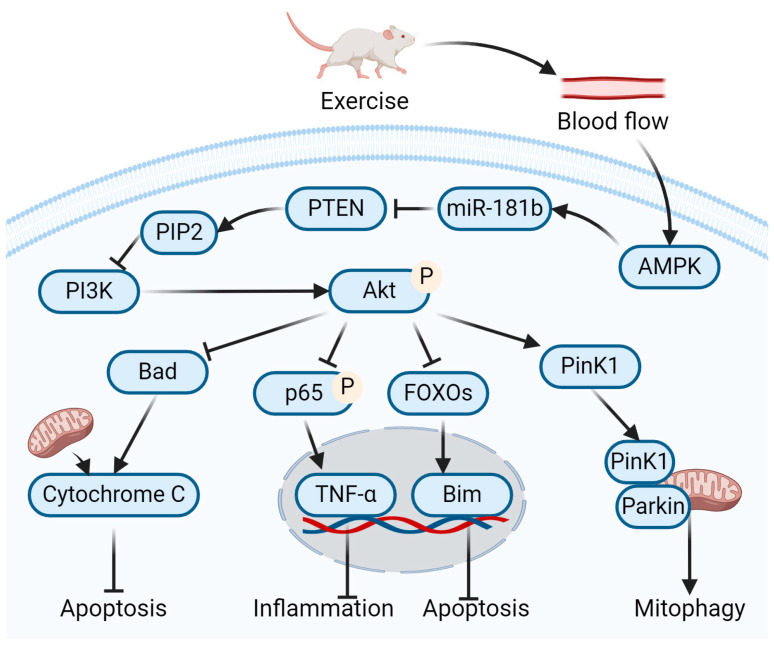
Role of miR-181b in exercise-mediated alleviation of DN. In DN animal models, exercise activates the AMPK signaling pathway in the vascular endothelial cell of the kidney by enhancing renal vascular blood flow. Activation of AMPK promotes Pink1 expression, leading to mitochondrial autophagy and a reduction in oxidative stress. AMPK activation also increases the expression of miR-181b, which inhibits phosphatase and tension homolog (PTEN)-mediated conversion of PIP3 into PIP2, thereby enhancing AKT phosphorylation. It plays an anti-inflammatory and anti-apoptotic role by inhibiting BAD signaling pathway, the phosphorylation of P65, and the nuclear translocation of FOXOs. At the same time, it can promote mitophagy by activating Pink1 signaling pathway. Created with BioRender.com, accessed on 12 March 2024.

**Figure 3 ijms-25-03605-f003:**
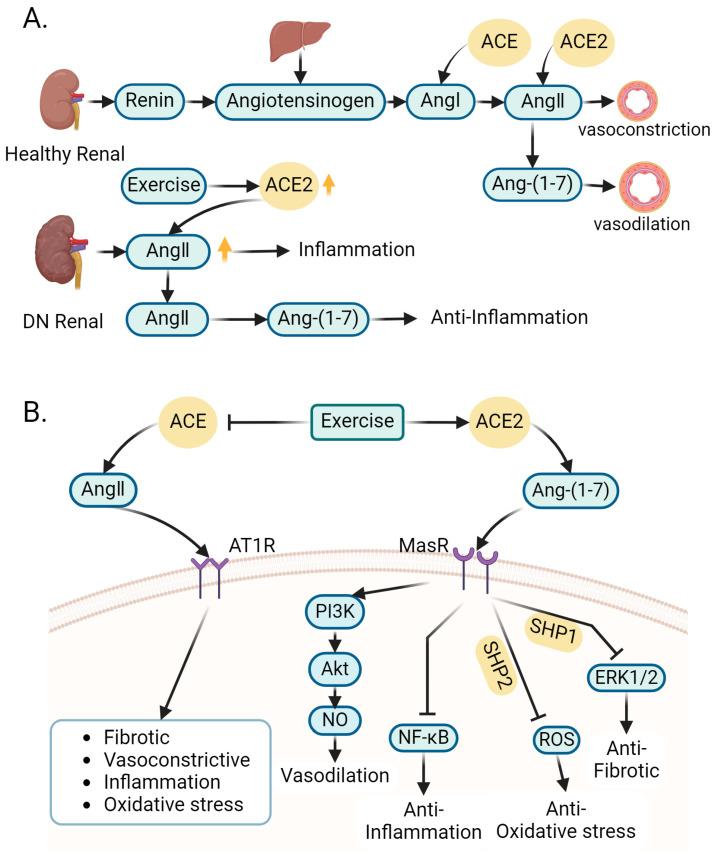
Role of exercise-regulated renin–angiotensin system (RAS) in amelioration of DN. (**A**) Development of diabetes leads to overactivation of the RAS in the kidney. ACE converts Ang I (secreted by the liver) into Ang II. Ang II, by binding to its receptor AT1R, has the effect of constricting blood vessels. The overactivation of Ang II/AT1R signaling can cause inflammation. ACE2 has antagonistic effects against ACE, which can degrade Ang II to produce the vasodilatory heptapeptide Ang-(1-7), an essential factor for maintaining renal function homeostasis. Upward yellow arrows mean up-regulation of its activity. (**B**) Exercise can decrease ACE activity and increase ACE2 activity. Increased ACE2 activity leads to high level of Ang-(1-7), whose binding to its receptor (MasR) results in effects opposite to Ang II. It inhibits ERK1/2 signaling pathways and ROS generation through SH2-containing protein tyrosine phos-phatase-1 (SHP1) and SHP2 of the Src family, and it also inhibits NF-κB signaling pathways and increases NO production, which leads to actions such as vasodilation and anti-fibrotic effects. Created with BioRender.com, accessed on 12 March 2024.

**Figure 4 ijms-25-03605-f004:**
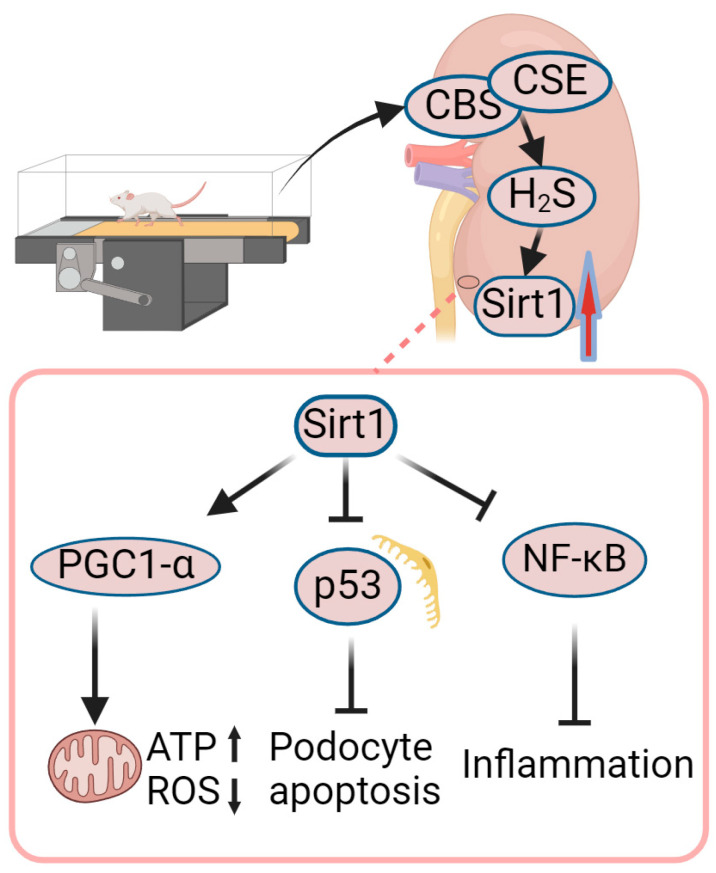
Role of exercise-mediated increase in Sirt1 in amelioration of DN. Exercise training impedes the downregulation of cystathionine-β-synthase (CBS) and cystathionine-γ-lyase (CSE) and restores renal H2S production in DN mice, which enhances renal Sirt1 expression and is accompanied by inhibition of the p53-mediated pro-apoptotic pathway. At the same time, Sirt1 stimulates mitochondrial biogenesis through deacetylation and activation of PGC1-α. Sirt1 can also exert anti-inflammatory effects by deacetylating the p65 subunit, thereby inhibiting the activation of the NF-κB signaling pathway. The upward red arrow means up-regulation of expression. Created with BioRender.com, accessed on 12 March 2024.

**Figure 5 ijms-25-03605-f005:**
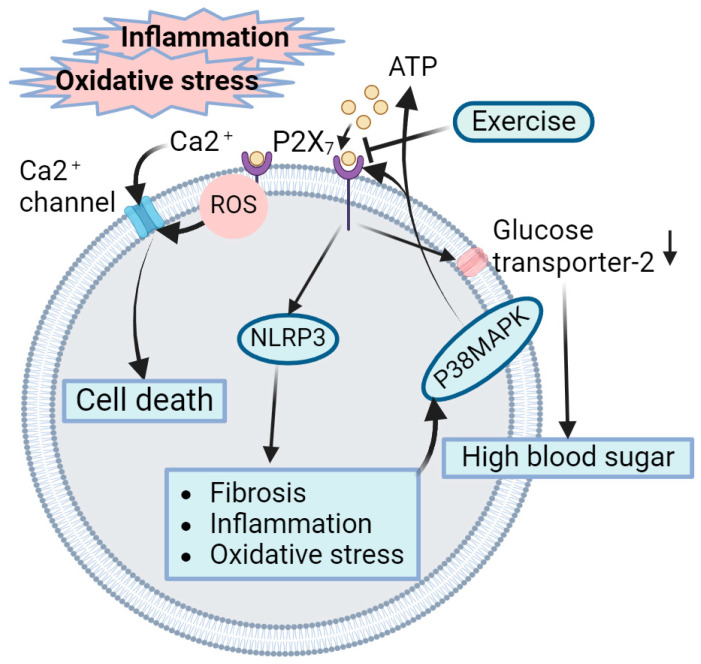
Role of exercise-mediated inhibition of P2X7 receptors in amelioration of DN. ATP accumulates at sites of tissues with damage and inflammation. ATP acts by activating and binding to P2X7R gated channels, which activates NLRP3 inflammasome in renal tubular epithelial cells. Intracellular ROS acts as an agonist, binding to the intracellular terminals of the P2X7R and opening the voltage-dependent Ca^2+^ channels, which causes cell death. P2X7R activation can also down-regulate the activity of glucose transporter-2 on the plasma membrane of kidney cells, which can promote uncontrolled high blood sugar. In the meantime, oxidative stress and hyperglycemia can activate p38 MAPK signaling to increase extracellular ATP levels and P2X7R expression, which forms a vicious circle. Exercise can reduce the activation and expression of P2X7R in renal tubular epithelium to exert a protective role against DN. The downward arrow on the right of the Glucose transporter-2 means a down-regulation of its activity. Created with BioRender.com, accessed on 12 March 2024.

**Figure 6 ijms-25-03605-f006:**
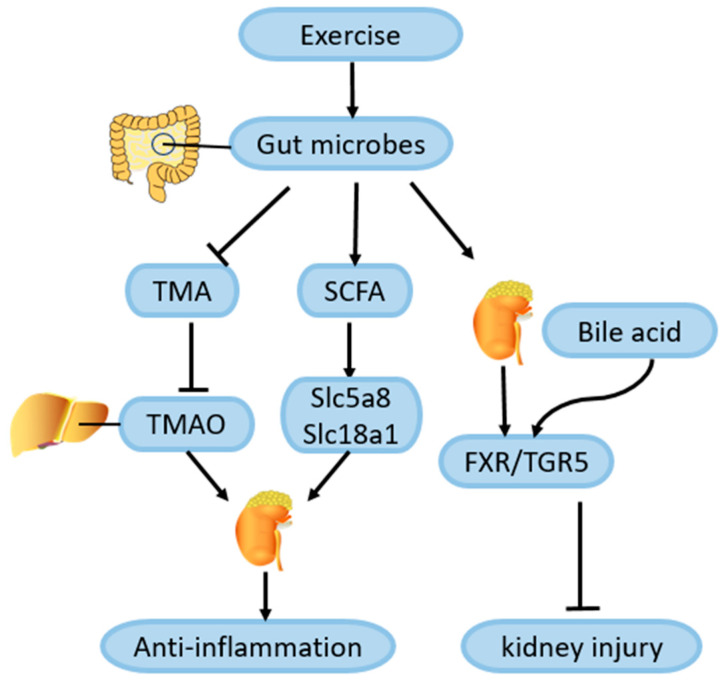
Role of exercise-mediated gut microbiota changes in amelioration of DN. The effects of exercise on intestinal microbes have been shown to increase the biosynthesis of short-chain fatty acids (SCFAs), which enter the circulation to relieve kidney inflammation by acting on transporters such as Slc5a8 and Slc18a1 in inflammatory cells and renal tubular epithelial cells. Exercise may also reduce the production of trimethylamine (TMA) in the intestine, and then the oxidation of TMA to trimethylamine-n-oxide (TMAO) in the liver is reduced, thereby reducing the circulation level of TMAO and alleviating renal inflammation. In addition, the changes in intestinal microbes caused by exercise may also promote bile acid absorption in the gut to activate the signaling of bile acid receptor FXR/TGR5 in the kidney to alleviate renal injury, the mechanism of which remains to be further studied. Icons are from ScienceSlides of Visiscienece.

**Figure 7 ijms-25-03605-f007:**
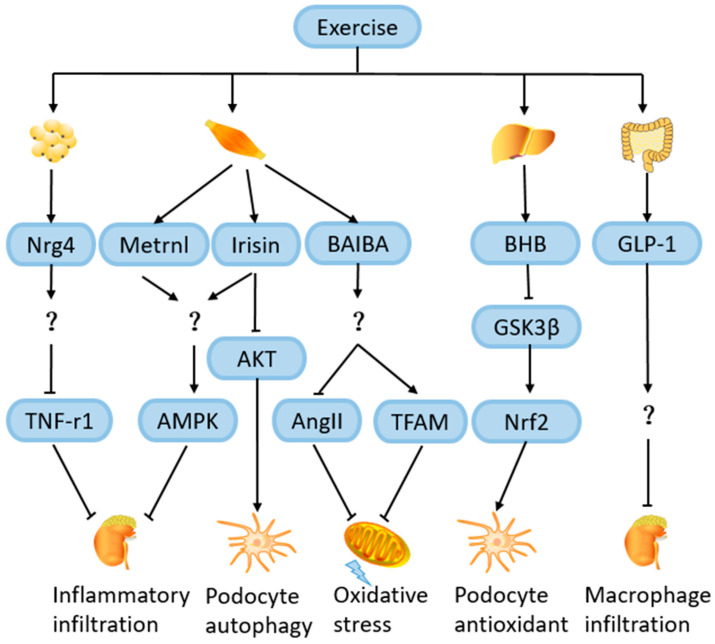
Role of exercise-induced hormones and metabolites in the protection against DN. After exercise, adipose tissue releases neuregulin 4 (Nrg4) to inhibit TNF receptor 1 (TNF-r1) in the kidney and skeletal muscle releases Irisin and Metrnl, activating AMPK signaling pathway and inhibiting AKT signaling pathway in the kidney. These effects above play an anti-inflammatory role in the kidney. In addition, exercise-induced release of BAIBA from the muscle activates transcription factor A mitochondria (TFAM) in the kidney and blocks the action of Ang II, thus promoting mitochondrial biogenesis in the kidney and inhibiting oxidative stress. Moreover, β-hydroxybutyrate (BHB) released from the liver after exercise can promote the accumulation of Nrf2 in the nucleus by inhibiting glycogen synthase kinase 3β (GSK3β), thus reducing the damage to podocytes. Exercise also promotes the release of Glucagon-like peptide 1 (GLP-1) from the gut, which somehow reduces macrophage infiltration in the kidney. The mechanism by which these exercise-induced hormones and metabolites protect against DN needs further study. The question mark means that the mechanism is not yet clear. Icons are from ScienceSlides of Visiscienece.

## Data Availability

No new data were created.

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
