# Peer review of "Exercise in Diabetic Nephropathy: Protective Effects and Molecular Mechanism"

_ijms, 2024, doi:10.3390/ijms25073605_

Round 1

Reviewer 1 Report

Comments and Suggestions for Authors

all the citations in the text, when mentioned with the author name, must be cited correctly. (e. g. 110, 115, 139,  etc). 

Author Response

Comments #1:

Comments 1: all the citations in the text, when mentioned with the author name, must be cited correctly. (e. g. 110, 115, 139, etc).
Answer:

Thank you for reviewing our manuscript and offering valuable advice. In the revised manuscript, all the author's name cited in the article has been changed to the family name. Please see these changes that are highlighted in yellow in the revised manuscript.

Reviewer 2 Report

Comments and Suggestions for Authors

This article is a review article that summarizes clinical research and basic research examining the mechanism of the effectiveness of exercise therapy in diabetic nephropathy. This review uses figures and tables effectively to provide easy-to-understand commentary. However, in my opinion, there are some points that should be corrected in the way of writing:

1.     I think that only the family name is sufficient for the author's name cited in the text.

2.     The references number in the references section is duplicated.

Author Response

Comments #1:

I think that only the family name is sufficient for the author's name cited in the text.
Answer:

We appreciate the reviewer’s suggestion. In the revised manuscript, all the author's name cited in the text has been changed to the family name. Please see these changes that are highlighted in yellow in the revised manuscript.

Comments #2:

The references number in the references section is duplicated.
Answer:

We thank the reviewer for pointing out our mistake. The references number has been corrected in the revised manuscript. Please let us know if there still exist some problems, we will continue to revise.

Reviewer 3 Report

Comments and Suggestions for Authors

Page 2, line 72: change "studies" into "participants"

Page 2, lines 84-86: there are 2 sentences with the same meaning; remove one of them

Page 3, line 115: Victor M are both given names; give family name instead (it should be Baiao VM)

Page 3, lines 127-131: the Authors cite a study od Ana Carolina et al, but there is no such paper listed in References

Page 3, line 149: change "Mai Ots" into "Pechter et al"

Page 4, line 156: insert comma, instead of a dot

Page 5, line 188: "Souza, C.S" - remove comma and a dot

Page 11, line 432: change "is" into "are"

Page 11, lines 440-445: cited paper of Toni M et al is lacking in References

Page 13, line 546: Betsy C [120] are both given names; give family name of the paper's first author instead

Page 14, line 563: Melissa [124] is a given name; give family name instead

Page 15, line 602: Guilherme [62] is a given name; give family name instead

Page 17, line 667: Irena [144] is a given name; give family name instead

Page 17, line 686: Agata [151] is a given name; give family name instead

Author Response

Comments #1:

Page 2, line 72: change "studies" into "participants".

Answer:

We thank the reviewer’s comment. We have changed “studies” into “participants” in page 2, line 72. Please see this change that is highlighted in yellow in the revised manuscript.

Comments #2:

Page 2, lines 84-86: there are 2 sentences with the same meaning; remove one of them.

Answer:

We thank the reviewer’s suggestion. We have removed the sentence “Exercise can improve cardiorespiratory fitness.” in Page 2, lines 84-85. The reserved sentence was highlighted in yellow. Please see this change that is highlighted in yellow in the revised manuscript.

Comments #3:

Page 3, line 115: Victor M are both given names; give family name instead (it should be Baiao VM).

Answer:

We thank the reviewer’s suggestion. We have changed the name of the author from “Victor M” to “Baiao” in Page 3, line 118. Please see this change that is highlighted in yellow in the revised manuscript.

Comments #4:

Page 3, lines 127-131: the Authors cite a study od Ana Carolina et al, but there is no such paper listed in References.

Answer:

Thank you for advising valuable advice. We apologize for forgetting to cite this paper. The citation here should be “Physical exercise as a friend not a foe in acute kidney diseases through immune system modulation”. DOI: 10.3389/fimmu.2023.1212163. The reference number of this paper was highlighted in yellow in Page 3, lines 130-131. Please see this change that is highlighted in yellow in the revised manuscript.

Comments #5:

Page 3, line 149: change "Mai Ots" into "Pechter et al".

Answer:

We thank the reviewer’s suggestion. We have changed the name of the author from “Mai Ots” to “Pechter” in Page 3, line 149. Please see this change that is highlighted in yellow in the revised manuscript.

Comments #6:

Page 4, line 156: insert comma, instead of a dot

Answer:

We thank the reviewer’s suggestion. We have inserted a comma instead of a dot in Page 4, line 159. Please see this change that is highlighted in yellow in the revised manuscript.

Comments #7:

Page 5, line 188: "Souza, C.S" - remove comma and a dot

Answer:

We thank the reviewer’s suggestion. We have removed the comma and the dot in "Souza, C.S" in Page 5, line 191, and changed it to “Souza”. Please see this change that is highlighted in yellow in the revised manuscript.

Comments #8:

Page 11, line 432: change "is" into "are"

Answer:

We appreciate the reviewer’s suggestion. We have changed “is” into “are” in Page 11, line 432. Please see this change that is highlighted in yellow in the revised manuscript.

Comments #9:

Page 11, lines 440-445: cited paper of Toni M et al is lacking in References

Answer:

Thank you for advising valuable advice. We apologize for forgetting to cite this paper. The citation here should be “Shear stress mediates endothelial adaptations to exercise training in humans”. DOI: 10.1161/HYPERTENSIONAHA.109.146282. The reference number of this paper was highlighted in yellow in Page 11, lines 440-441. Please see this change that is highlighted in yellow in the revised manuscript.

Comments #10:

Page 13, line 546: Betsy C [120] are both given names; give family name of the paper's first author instead

Answer:

We thank the reviewer’s suggestion. We have changed the name of the author from “Betsy C” to “Wertheim” in Page 13, line 547. Please see this change that is highlighted in yellow in the revised manuscript.

Comments #11:

Page 14, line 563: Melissa [124] is a given name; give family name instead

Answer:

We thank the reviewer’s suggestion. We have changed the name of the author from “Melissa” to “Erickson” in Page 14, line 564. Please see this change that is highlighted in yellow in the revised manuscript.

Comments #12:

Page 15, line 602: Guilherme [62] is a given name; give family name instead

Answer:

We thank the reviewer’s suggestion. We have changed the name of the author from “Guilherme” to “Formigari” in Page 15, line 604. Please see this change that is highlighted in yellow in the revised manuscript.

Comments #13:

Page 17, line 667: Irena [144] is a given name; give family name instead

Answer:

We thank the reviewer’s suggestion. We have changed the name of the author from “Irena” to “Audzeyenka” in Page 17, line 668. Please see this change that is highlighted in yellow in the revised manuscript.

Comments #14:

Page 17, line 686: Agata [151] is a given name; give family name instead

Answer:

We thank the reviewer’s suggestion. We have changed the name of the author from “Agata” to “Winiarska” in Page 17, line 687. Please see this change that is highlighted in yellow in the revised manuscript.

Reviewer 4 Report

Comments and Suggestions for Authors

There are many different opinions about the mechanism of diabetic nephropathy, and several methods for treating it are known. Among them, this paper mentions the effects of exercise and explains the related mechanisms, providing a meaningful summary based on various evidence.

Minor points

1. In lines 36-38, the authors should provide the references you categorized.

2. In lines 99-101, the authors should define the intensity of exercise.

Author Response

Comments #1:

In lines 36-38, the authors should provide the references you categorized.

Answer:

We appreciate the reviewer’s comment. The reference is “Update on diabetic nephropathy: core curriculum 2018” by Umanath et al., which has been provided and highlighted in yellow in line 38. Please see this change that is highlighted in yellow in the revised manuscript.

Comments #2:

In lines 99-101, the authors should define the intensity of exercise.

Answer:

We appreciate the reviewer’s comment. We restated the content of the paper in lines 98-103, in which the definition of exercise intensity is “they classified physical and recreational activities as light, mixed and vigorous based on calories burned per minute (5, 7.5 and 10 kilocalories per minute)”. Please see this change that is highlighted in yellow in the revised manuscript.